# FAPI PET versus FDG PET, CT or MRI for Staging Pancreatic-, Gastric- and Cholangiocarcinoma: Systematic Review and Head-to-Head Comparisons of Diagnostic Performances

**DOI:** 10.3390/diagnostics12081958

**Published:** 2022-08-12

**Authors:** Sophie E. M. Veldhuijzen van Zanten, Kay J. Pieterman, Bas P. L. Wijnhoven, Ilanah J. Pruis, Bas Groot Koerkamp, Lydi M. J. W. van Driel, Frederik A. Verburg, Maarten G. J. Thomeer

**Affiliations:** 1Department of Radiology and Nuclear Medicine, Erasmus Medical Center, Dr. Molewaterplein 40, 3015 GD Rotterdam, The Netherlands; 2Department of Surgery, Erasmus Medical Center, 3015 GD Rotterdam, The Netherlands; 3Department of Gastroenterology and Hepatology, Erasmus Medical Center, 3015 GD Rotterdam, The Netherlands

**Keywords:** FAPI PET, diagnostic performance, pancreatic carcinoma, gastric carcinoma, cholangiocarcinoma

## Abstract

Introduction: There is a pressing demand for the development of cancer-specific diagnostic imaging tools, particularly for staging of pancreatic-, gastric- or cholangiocarcinoma, as current diagnostic imaging techniques, including CT, MRI and PET using FDG, are not fully adequate. The novel PET-tracer “FAPI” has the potential to visualize even small tumour deposits employing the tumour-specific expression of fibroblast-activating protein (FAP) in malignant cells. Methods: We performed a systematic review to select studies investigating the use of FAPI PET for staging pancreatic-, gastric- and cholangiocarcinoma (PROSPERO CRD42022329512). Patient-wise and lesion-wise comparisons were performed for primary tumour (T), lymph nodes (N), organ metastases (M) and peritoneal carcinomatosis (PC). Maximum standardized uptake values (SUVmax) and tumour-to-background ratios (TBR) were compared between PET using FAPI versus FDG (if reported). Results: Ten articles met the inclusion criteria. In all studies, FAPI PET showed superiority over FDG-PET/CT/MRI for the detection of T, N, M and PC, both in the patient-wise and in lesion-wise comparisons (when performed). Additionally, higher SUVmax and TBRmax values were reported for use of FAPI compared to FDG. Conclusions: The positive results of this review warrant prospective clinical studies to investigate the accuracy and clinical value of FAPI PET for diagnosing and staging patients with pancreatic-, gastric- and cholangiocarcinoma.

## 1. Introduction

The use of sophisticated imaging techniques, such as contrast-enhanced computed tomography (hereafter referred to as CECT), magnetic resonance imaging (MRI) and positron emission tomography (PET) has become the standard of care in the diagnostic work-up and follow-up of cancer patients. For a wide range of indications, PET employing a [18F]-2-Fluoro-Deoxy-Glucose (FDG) tracer enables the detection of increased glycolysis present in cancer cells, even in the absence of—or preceding—structural abnormalities. However, as glucose metabolism is ubiquitous in the human body, especially smaller lesions may suffer from insufficient lesion-to-background signal ratios, limiting the sensitivity of FDG. Additionally, as FDG not only accumulates in cancer cells but likewise in areas of infection and inflammation, specificity is limited.

Particularly in preoperative staging of a potentially curable pancreatic-, gastric- or cholangiocarcinoma, accurate imaging plays a crucial role. In contrast to other abdominal tumours, including colon carcinoma and ovarian carcinoma, positive distant lymph nodes, organ metastases or peritoneal/omental implants preclude any surgical resection. In these cases, patients will be offered alternative therapies, including palliative and supportive care. Currently, ubiquitously available imaging techniques, including FDG PET, CECT and MRI, are known to be largely unsatisfactory for this indication. For this reason, surgical exploration by laparoscopy is performed prior to eventual curative surgery in order to detect the possible localisations of tumour spread that are under detected by imaging. The yield of this invasive procedure, however, is low (i.e., between 14% and 17%) with moderate sensitivity of 71% [1,2,3]. The underestimation of local tumour status may result in futile surgeries when patients are found to be incurable only during intended curative surgery. Overestimation, on the other hand, prevents patients from receiving a potential curative treatment.

Recently, it was discovered that cancer-associated fibroblasts show a high expression of fibroblast activation protein (FAP), while expression levels in normal human tissues are generally very low [4]. This pathophysiological process can be imaged by PET, using the novel ligand “fibroblast activation protein inhibitor (FAPI)” labelled to fluorine-18 or gallium-68. First clinical studies show that FAPI PET is safe and may outperform the use of FDG [5]. In a study including 80 patients with 28 types of cancer, FAPI PET yielded better tumour-to-background signal ratios (TBR), higher sensitivity for the detection of new and recurrent tumour locations, and better discrimination from benign lesions and acute inflammation [6]. The aim of this study was to evaluate the possible yield of FAPI PET (combined with either (CE)CT or MRI), compared to conventional diagnostic techniques, including FDG PET, CECT and MRI, either as a stand-alone or combined, for the staging of pancreatic-, gastric- and cholangiocarcinoma. This was achieved by performing a systematic literature review and the patient-wise and lesion-wise head-to-head comparisons of the diagnostic performance of these techniques.

## 2. Materials and Methods

This systematic review adheres to the international PRISMA-DTA guidelines for Systematic Reviews and Meta-Analysis of Diagnostic Test [7]. The research protocol was registered in PROSPERO (ID nr. CRD42022329512) [8].

### 2.1. Search of Literature

A systematic search of current literature on FAPI PET was performed initially in June 2021 and updated in April 2022. A biomedical information specialist was consulted to ensure that all relevant keywords and synonyms were included in the search. Precise search terms can be found in the Appendix A section. Keywords included synonyms of “FAPI PET”, “FDG PET”, “(CE)CT” and “MRI” in combination with synonyms for pancreatic-, gastric- or cholangiocarcinoma. Embase, Medline, Web of Science and Cochrane library were scanned for relevant articles.

### 2.2. Study Eligibility

This systematic review focused on diagnostic studies into pancreatic-, gastric- and cholangiocarcinoma, either as main focus or in combined investigations with other types of tumours. All subtypes of either of these cancer entities were considered eligible for analysis. Reviews and animal studies were excluded. Other exclusion criteria were as follows: the article was not written in English, the study did not focus on the diagnostic tests of interest (i.e., FAPI PET, FDG PET, (CE)CT MRI) and the study did not report original numbers to calculate true positives, false positives, true negatives and false negatives for either of the diseases individually.

### 2.3. Quality Assessment

All papers were screened by SV and KP independently for methodological quality using the QUADAS-2 assessment tool [9]. All potentially relevant articles were scored on methodological aspects in the domains of participant selection, index test, reference standard and participant flow and timing. In case of disagreement, a re-reading by both reviewers was performed to reach consensus on the final appointed QUADAS score.

### 2.4. Data Items and Analysis

In the subsequent full text analysis of the selected articles, the following study and patient characteristics were retrieved if available: year of publication, study design (prospective or retrospective), the number of participants, and gender and age of the patients. Per article, for each tumour type of interest (i.e., pancreatic-, gastric- or cholangiocarcinoma), the number of detected lesions by FAPI PET and FDG PET and/or (CE)CT and or MRI, were extracted for primary tumour, lymph nodes, organ metastases and peritoneal carcinomatosis (i.e., the presence of peritoneal/omental tumour depositions, denoted as T/N/M/PC, respectively). If sufficient detailed information was available, both patient-wise and lesion-wise comparisons were performed. For patient-wise analysis, the number of patients with positive outcomes by FAPI PET and by FDG PET and/or (CE)CT and or MRI were registered. For lesion-wise analysis, the number of detected lesions by each of the imaging techniques were registered for all of the study parameters, T/N/M/PC, individually. Additionally, the used reference standard (e.g., histological proof or follow-up imaging >3 months to confirm the malignant nature of the lesions) was noted when reported. If pathological proof or follow-up imaging was available, sensitivity was calculated accordingly. The methodology was chosen in such manner that a meta-analysis could be performed in case of sufficient data quality.

In addition, maximum standardized uptake values (SUVmax) and maximum target to background ratios (TBRmax) were extracted from each article and used for both patient-wise and lesion-wise comparisons between FAPI PET and FDG PET/CT/MRI.

## 3. Results

The literature search resulted in 40 studies, of which titles and abstracts were screened for relevance. In accordance, 18 studies were excluded for not matching the required study type (clinical images (*n* = 7), editorial letters (*n* = 2), poster presentations (*n* = 5), technical papers or animal studies (*n* = 4)). Another nine were excluded because they did not meet the relevance criteria (i.e., studies not reporting FAPI PET results for pancreatic-, gastric- or cholangiocarcinoma) and three for not reporting numeric data. Finally, ten articles met all inclusion criteria and were further analysed [10,11,12,13,14,15,16,17,18,19]. Figure 1 shows a flowchart of the process of in- and exclusion.

### 3.1. Study and Patient Characteristics

Study characteristics are summarized in Table 1. The included studies were all published in 2021 and encompassed five prospective and five retrospective studies. Three studies (indicated by #) performed post hoc retrospective analyses on sub-cohorts of patients from a prospective parent study, which was performed between July 2019 and March 2020 by Chen, H., et al. (2020, clinicaltrials NCT04416165). The subsequent studies by Guo, Pang and Zhao et al., however, included more patients on the specific diseases of interest to us. The inclusion periods of these studies differed by a total of 6 and 8 months from the initial study, respectively. Guo et al. performed an in-depth analysis of 12 patients with intrahepatic cholangiocarcinoma. Of these patients, four may have also been included in the parent study (indicated between brackets). The two studies published by Shi et al. also made use of a corresponding IRB protocol (#ZS1810). However, based on the comparative imaging techniques used in each of these studies (i.e., FDG PET in [15] versus CECT/MRI in [16]) no overlap of patients was expected.

Per study, a median total of 27 patients were included (range 13–68), of which only a few (indicated by >) suffered from the cancer types of our specific interest. The median age of patients was 60 years (range 24–87 years). Six studies compared the use of FAPI PET-CT with FDG PET-CT; one study compared FAPI PET-CT with FDG PET-CT, CECT and MRI; two studies compared FAPI PET-CT with CECT (of which one also investigated the use of MRI); and one study compared FAPI PET-MRI with FDG PET-CT. The radiopharmaceuticals that were used were [^68^Ga]Ga-DOTA-FAPI-04 (in short ^68^Ga-FAPI-04; nine studies) and [^68^Ga]Ga-DOTA-FAPI-46 (in short ^68^Ga-FAPI-46; one study).

### 3.2. Quality Assessment

The results of methodological quality screening using QUADAS-2 are shown in Table 2. Within the evaluated domains of all ten studies, there were four bias concerns. In two studies the risk of bias with regards to the participant selection (column 3) was considered high, because patients were enrolled only when findings of prior FDG PET were negative or inconclusive. This likely introduced bias in relation to the head-to-head comparison of performance between FAPI and FDG, particularly in terms of sensitivity and specificity. Possible selection bias for the above-described retrospective post hoc studies was classified as “unclear”.

In three studies the risk of bias with regards to the reference standard (column 5) was considered high for metastatic lesions (not for primary tumour), because the pathological confirmation or imaging follow-up (>3 months) of some of the detected lesions was missing. In the other QUADAS-2 domains of the studies, no certain or significant risks of bias were found.

In all studies that evaluated primary tumour detectability using FAPI and FDG (*n* = 9), histological proof or follow-up imaging (>3 months) was obtained (Table 3). For lymph node metastases, organ metastases and peritoneal/omental tumour depositions, however, the use of a reference standard was not always reported or not specifically investigated. In some studies, the detection rate of metastatic lesions (N/M/PC) was only compared between FAPI PET and FDG PET/CT/MRI, without use of a reference standard. In others, not all metastatic lesions (N/M/PC) were verified by a reference standard, probably because in most cases a large number of FAPI/FDG positive metastatic lesions were found, since most patients were included during advanced disease stages (which is not further reported upon).

### 3.3. Patient-Wise Analysis

Overall, FAPI PET showed a better detection rate compared to FDG PET/CT/MRI for pancreatic-, gastric- and cholangiocarcinoma (Table 4): with use of FAPI PET, combined with either CT or MR, primary tumours were visualized in all patients (column 4), while less frequently with FDG PET/CT/MRI (column 5). With regards to metastases, FAPI also showed better detection rates over the use of FDG for the identification of affected lymph nodes, organ metastases and peritoneal/omental metastases (column 4 versus 5), although two studies (investigating pancreatic- and cholangiocarcinoma, respectively) did not verify these results by the use of a reference standard (i.e., histological proof or follow-up imaging (>3 months) to confirm the malignant nature of the lesions).

### 3.4. Lesion-Wise Analysis

In all studies, more lymph node-, organ- and peritoneal/omental metastases were detected by FAPI PET (column 6), compared to FDG PET (column 7), regardless of the additional use of CT or MRI.

### 3.5. Sensitivity of FAPI PET versus FDG PET/CECT/MRI

Appendix A shows the sensitivity analyses (mostly available for patient-wise analysis) of all studies in which histological confirmation or follow-up imaging was performed. Numbers were either reported in the original manuscript or calculated based on published raw data. Results show the overall higher sensitivity of FAPI PET (85–100%; column 4), compared to FDG PET/CT/MRI (ranging from 36–73%, column 5).

### 3.6. Additional Findings—True Positive (i.e., Also Malignant)

Shi et al. [15] reported increased uptake of FAPI at the right lobe of the thyroid in one patient, which turned out to be a papillary carcinoma. This lesion was not detected by FDG PET-CT.

### 3.7. Additional Findings—False Positive (i.e., Avid—Non-Malignant)

Diffuse high uptake in pancreatitis was observed by Chen [10] (*n* = 1), Rohrich [14] (*n* = 8/19) and Pang et al. [18] (*n* = 12/26). In the latter two studies, pancreatic carcinoma could be differentiated from underlying pancreatitis by increasing/stable vs. decreasing uptake over time (measured at 10, 60 and 180 min and 1 h and 3 h post-injection, respectively).

Shi et al. [16] reported significantly increased FAPI uptake in cirrhotic (*n* = 7) versus non-cirrhotic (*n* = 8) liver parenchyma (SUVmean 1.39 vs. 0.46), which was, however, still significantly lower than the uptake observed in hepatocellular carcinoma and intra-hepatic cholangiocarcinoma (SUVmax 13.6). These findings correlated with the results of Guo et al. [11] (SUVmean 4.84 +/− 1.64 vs. 1.99 +/− 0.65) and Pang et al. [12] (not further specified). Chen et al. [10] also detected high FAPI uptake in liver at sites of post-treatment (i.e., partial hepatectomy) inflammation (*n* = 2); similarly, Guo et al. [11] found post-treatment inflammatory reaction (*n* = 4) and inflammatory granuloma (*n* = 1) in liver, as well as positive FAPI uptake in pulmonary infection (*n* = 1) and thyroid adenoma (*n* = 1). Chen [10] and Pang [12] et al. furthermore reported intense (not further specified) FAPI uptake in granulomatous lesions from tuberculosis (in brain, lung and lumbar vertebra). Other false positive lesions were related to chronic inflammatory or reactive processes in joints, tendons, scars, mastitis and around a y-nail implant [14]. Other studies found cases with myelofibrosis [12], a corpus luteum cyst [16] and a vertebral fracture [15].

### 3.8. Additional Findings—True Negative (i.e., Non Avid—Non Malignant)

Benign lesions that did not show significant uptake on FAPI PET (resembling or non-significantly differing from FDG uptake) were: esogastritis (*n* = 2), pancreatic cystadenoma (*n* = 1), hepatic adenoma (*n* = 1) [10], benign nodules (not further specified, *n* = 2) [11], haemangiomas and granulomas [15]. In contrast Pang et al. [12] reported false positive results for esogastritis and abscess.

### 3.9. False-Negative Results of FAPI PET

Chen et al. [10] reported on three patients (out of 68) with both FAPI and FDG PET undetected malignant lesions: two with lung adenocarcinoma (both presenting with a solitary pulmonary nodule) and one with renal clear-cell carcinoma.

### 3.10. Tracer Visualization

In all studies, both the SUVmax as well as the TBRmax were markedly higher for primary tumours, lymph nodes, and organ- and peritoneal/omental metastases when FAPI PET was used compared to FDG PET (Appendix A).

## 4. Discussion

This study investigates the diagnostic yield of the novel FAPI PET technique, combined with either CT or MRI, by means of a systematic review and intra-patient and per-lesion head-to-head comparisons with conventional diagnostic techniques, including CT and MRI, either as stand-alone or combined with FDG PET in pancreatic-, gastric- and cholangiocarcinoma. Despite the small number of studies that yet have been conducted, our results point towards a clear benefit of FAPI PET compared to the currently prevailing diagnostic techniques.

The results of our literature review show that FAPI PET was able to detect more patients with primary disease and/or nodal-, organ- and peritoneal/omental metastases. Secondly, a larger number of lesions per patient could be identified using FAPI PET, thereby providing a better estimation of disease extension and tumour load. This indicates that FAPI PET is a promising technique for diagnosing and staging pancreatic-, gastric- and cholangiocarcinoma and warrants prospective studies in larger patient groups to enable the further quantification of its clinical value and cost-effectiveness.

Other than preoperative staging, FAPI PET may also prove to be of value for staging before and during chemotherapy, in that it could serve as a valuable novel imaging biomarker for the follow-up of treatment response, in addition to use of the currently prevailing Response Evaluation Criteria in Solid Tumours (RECIST). Our head-to-head comparisons show that FAPI PET, compared to FDG PET, consistently provides higher SUVmax and TBRmax values for tumour detection. This is particularly beneficial for the detection of smaller lesions and lesions located in close proximity to abdominal structures with physiological tracer uptake (i.e., background signal). The highest physiological uptake of FAPI in the abdomen was found in kidneys, pancreas, spleen and colon, with SUV values equalling those of FDG [20]. Physiological FAPI uptake in the liver was generally lower compared to FDG uptake, which increases the likelihood of detecting liver metastases. Although specificity was not specifically investigated in this study, there are reports of false-positive findings on FAPI PET imaging. Examples include Schmorl nodes in bone [21], thyroiditis [22], haemangioma and pneumonia [23], solitary fibrous tumours [24] or chronic cholecystitis (i.e., chronic inflammation) [25,26]. Most of these lesions, however, would not be confused with a potential primary or metastatic lesion of the three tumour types studied here. Furthermore, also in patients undergoing radiation therapy, FAPI PET could be of benefit as FDG PET is not well-suited for differentiation between post-radiation acute inflammation and residual malignant disease. Whilst FAPI PET will show a high uptake in chronic inflammation, it generally does not show a significant uptake in acute inflammatory processes, thus allowing for the easier differentiation form true vital tumour remnants. However, to sufficiently prove this and subsequently implement FAPI PET in routine clinical practice, prospective well-designed phase 1, 2 and 3 trials need to evaluate its diagnostic performance for the indications reviewed in this work.

With respect to the other imaging techniques investigated; Guo et al. [11] reported the equal sensitivity of FAPI PET, CECT and MRI in patient-wise analyses, which all outperformed FDG PET, the better performance of MRI compared to all other imaging techniques for detection of intrahepatic (primary and metastatic) lesions specifically. Qin et al. reported enhanced confidence for the diagnosis of ovarian metastases (*n* = 6/14), hepatic metastases (*n* = 3) and uterine/rectal metastases (*n* = 1) with use of MRI (combined with FAPI PET) compared with FDG PET-CT in patients with gastric carcinoma. Sensitivities for comparative CECT used by Rohrich et al. [14], CECT/MRI by Shi et al. [16] and CECT used by Pang et al. [18] were not specifically reported.

With regards to novel therapeutic strategies for pancreatic-, gastric- and cholangiocarcinoma, FAPI also offers theranostic potential, meaning that its ligand can be bound to either diagnostic or therapeutic radioisotopes. Thus, not only can the ligand be labelled with gallium-68 for imaging but also with other radioisotopes emitting therapeutic radiation, such as yttrium-90, lutetium-177, holmium-166 or terbium-161. A great deal of research, however, is still needed to define the optimal isotope for each disease anew as the different histologies of various cancers may sometimes favour one over the other. Here, studies in the preclinical arena are still necessary before even thinking of progressing towards trials in humans.

The several limitations of this systematic review need to be addressed. First, given the relative novelty of the FAPI tracer, only a small number of studies (*n* = 10) were available for review. Of these, some were performed by the same research group and possible overlap of patient data could not be ruled out completely based on the reported information in the manuscripts. Additionally, most studies included different subtypes of tumours. For example, intrahepatic as well as extrahepatic cholangiocarcinoma, gastric tubular adenocarcinoma and diffuse type signet-ring cell carcinoma. Additionally, there was a large variation in disease stage at which patients were included, as well as in treatments given to the patients. This could have influenced reported outcomes, for instance because of varying tumour sizes known to relate to detectability in PET studies. The resulting range, from treatment naive patients to patients that underwent re-staging after therapy, may introduce bias with regards to possible specific sensitivities/specificities for primary staging and restaging.

Furthermore, use of a reference standard (i.e., pathological confirmation or follow-up imaging) was lacking or incomplete in some studies, particularly for (all individual) metastases in patients with a large number of lesions. It was therefore not always assured that all detected lesions by FAPI PET, particularly those not detected by FDG PET, were true-positive lesions. Additionally, although most studies reported separate results for each investigated tumour type, the analyses of metastatic lesions were often performed for multiple tumour types combined. Furthermore, metastatic lesions in different anatomical locations (e.g., nodal, peritoneal/omental) were not always analysed separately, hampering detailed analysis and resulting in large heterogeneity in SUVmax and TBRmax values.

Notwithstanding these limitations, the results of our review and head-to-head comparisons indicate that the introduction of FAPI PET in the diagnostic work-up of patients with pancreatic-, gastric- or cholangiocarcinoma will likely result in better staging and treatment planning, thus fulfilling a high unmet need and adding great clinical value.

## 5. Conclusions

### Implications for Clinical Practice and Directions for Further Research

The results of this systematic review and patient-wise and lesion-wise comparisons clearly show the great potential that FAPI PET-CT/-MRI holds for diagnostic imaging in gastric-, pancreatic- and cholangiocarcinoma, allowing for more accurate staging and possibly better patient-tailored treatment strategies and decision making. Based on these promising results, FAPI PET/CT/MRI will likely supplement or replace FDG PET/CT/MRI and possibly even exploratory laparoscopy in the diagnostic work-up of the cancer types here investigated. However, larger studies with more extensive tissue sampling or imaging follow-up seem desirable for a better assessment of the sensitivity and specificity of this novel radiotracer. Additionally, apart from diagnostic accuracy, the benefits in terms of patient-relevant outcome measures, such as overall and progression-free survival and the eventual theragnostic use of FAPI PET, are still to be assessed. 

## Figures and Tables

**Figure 1 diagnostics-12-01958-f001:**
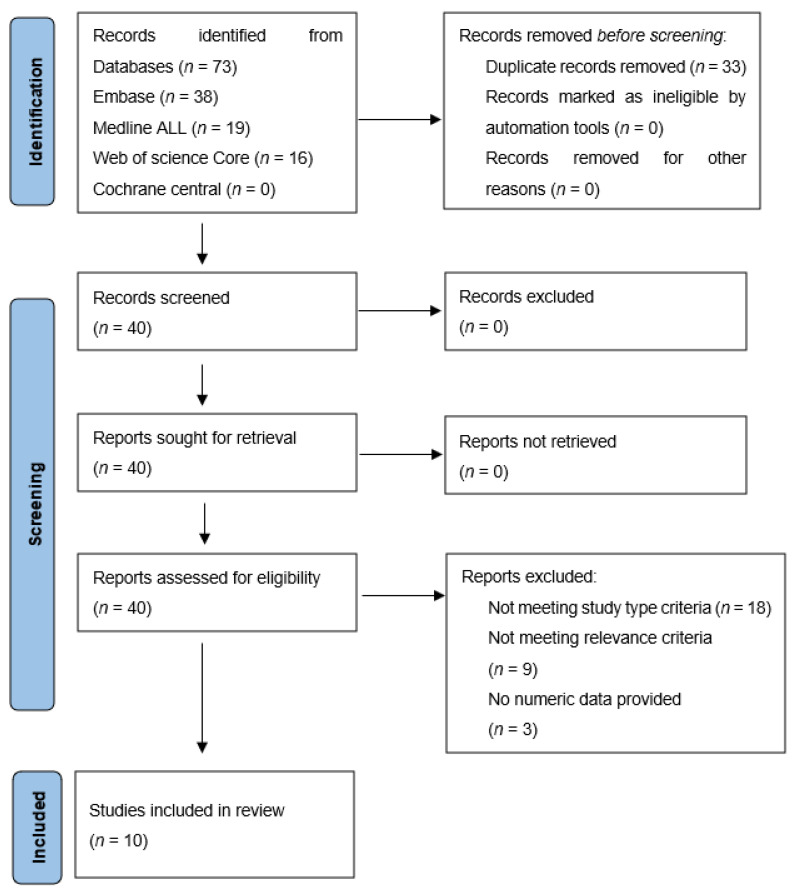
In- and exclusion of papers found by the literature search.

**Table 1 diagnostics-12-01958-t001:** Study characteristics.

Year	Author	Study Design	Period of pt. Inclusion	Cancer Type(s) Matching Our Interest	No. of pts. (Total > per Cancer Type of Interest Specifically)	Gender (M/F)	Median (Range) Age (in Years)	Modalities Studied	Radiopharmaceutical
**2020**	Chen, H. et al. [10]	Pro	July 2019 March 2020	P/G/Ch	68 > 1/12/8	40/28	57 (24–85)	FAPI PET-CT FDG PET-CT	^68^Ga-FAPI-04
**2021**	^#^ Guo, W. et al. [11]	Retro	October 2019 June 2020	Ch	34 > 12 (8)	25/9	61 (33–75)	FAPI PET-CT FDG PET-CT CECT MRI	^68^Ga-FAPI-04
**2021**	^#^ Pang, Y. et al. [12]	Retro	October 2019 June 2020	G	35 > 20 (12)	18/17	64 (53–68)	FAPI PET-CT FDG PET-CT	^68^Ga-FAPI-04
**2021**	^#^ Zhao, L. et al. [17]	Retro	October 2019 August 2020	P/ G	46 > 6 (1)/ 13 (12)	14/32	5 7(32–80)	FAPI PET-CT FDG PET-CT	^68^Ga-FAPI-04
**2021**	Qin, C. et al. [13]	Pro	June 2020 June 2020	G	20	9/11	56 (29–70)	FAPI PET MRI FDG PET-CT	^68^Ga-FAPI-04
**2021**	Rohrich, M. et al. [14]	Retro	not mentioned	P	19	10/9	64 (52–80)	FAPI PET CECT	^68^Ga-FAPI-04^68^ Ga-FAPI-46
**2021**	Shi, X. et al. [15]	Pro	January 2019 June 2020	Ch	20 > 3	18/2	58 (43–78)	FAPI PET-CT FDG PET-CT	^68^Ga-FAPI-04
**2021**	Shi, X. et al. [16]	Pro	January 2019 August 2020	G/Ch	17 > 1/2	13/3	63 (47–78)	FAPI PET-CT CECT/MRI	^68^GA-FAPI-04
**2021**	Pang, Y. et al. [18]	Retro	June 2020 January 2021	P	36 > 26	18/17	64 (53–68)	FAPI PET-CT FDG PET-CT CECT	^68^Ga-FAPI-04
**2021**	Kuten, J. et al. [19]	Pro	July 2020 December 2020	G	13	6/7	70 (35–87)	FAPI PET-CT FDG PET-CT	^68^Ga-FAPI-04

^#^ Retrospective studies based on patient (sub)cohorts generated in the prospective study by Chen, H. et al., 2020 [10]. Patients that may have been included in both the prospective parent study as well as in the subsequent retrospective studies were indicated between brackets. P = pancreatic carcinoma; G = gastric carcinoma; Ch = cholangiocarcinoma; Pro = prospective; Retro = retrospective; pt(s). = patient(s); no. = number; M/F = male/female; CECT = Contrast-enhanced CT; MRI = Magnetic Resonance Imaging; PET = positron emission tomography; FDG = fluorodeoxyglucose; FAPI = fibroblast activation protein inhibitor.

**Table 2 diagnostics-12-01958-t002:** QUADAS-2 analysis—subdomain scores (high/low/unclear) per study.

Year	Author	Risk of Bias—Participant Selection	Risk of Bias—Index Test	Risk of Bias—Reference Standard	Risk of Bias—Flow and Timing	Applicability Concerns—Participant Selection	Applicability Concerns—Index Test	Applicability Concerns—Reference Standard
**2020**	Chen, H. et al. [10]	high	low	low	low	low	low	low
**2021**	Guo, W. et al. [11]	unclear ^#^	low	low	low	low	low	low
**2021**	Pang, Y. et al. [12]	unclear ^#^	low	low	low	low	low	low
**2021**	Zhao, L. et al. [17]	unclear ^#^	low	low	low	low	low	low
**2021**	Qin, C. et al. [13]	low	low	low (primary tumour), high (metastasis)	low	low	low	low
**2021**	Rohrich, M. et al. [14]	low	low	Low (primary tumour), high (metastasis)	low	low	low	low
**2021**	Shi, X. et al. [15]	high	low	low (primary tumour) high (metastasis),	low	low	low	low
**2021**	Shi, X. et al. [16]	low	low	low	low	low	low	low
**2021**	Pang, Y. et al. [18]	low	low	low	low	low	low	low
**2021**	Kuten J. et al. [19]	low	low	low	low	low	low	low

^#^ Retrospective studies based on patient (sub)cohorts generated in the prospective study by Chen, H. et al., 2020 [10].

**Table 3 diagnostics-12-01958-t003:** Reported reference standards to confirm malignant nature of lesions.

Year	Author	Primary Tumour	Lymph Node Metastasis	Organ Metastasis	Peritoneal/Omental Metastasis
**2020**	Chen, H. et al. [10]	Pathology (*n* = 49/68)/ Imaging FU (*n* = 19)	Pathology/ Imaging FU	Pathology/ Imaging FU	Pathology/ Imaging FU
**2021**	Guo, W. et al. [11]	Pathology (all patients)	Pathology/ Imaging FU	Pathology/ Imaging FU	Pathology/ Imaging FU
**2021**	Pang, Y. et al. [12]	Pathology (all patients)	Pathology/ Imaging FU	Pathology/ Imaging FU	Pathology/ Imaging FU
**2021**	Zhao, L. et al. [17]	n.p.	n.p.	n.p.	Pathology (*n* = 27/46)/ Imaging FU (*n* = 19)
**2021**	Qin, C. et al. [13]	Pathology (all patients)	Pathology (*n* = 5/14)	Pathology (*n* = 5/14)	Pathology (*n* = 5/14)
**2021**	Rohrich, M. et al. [14]	Pathology (all patients)	-	-	-
**2021**	Shi, X. et al. [15]	Pathology (all patients)	-	-	-
**2021**	Shi, X. et al. [16]	Pathology (all patients)	n.p.	n.p.	n.p.
**2021**	Pang, Y. et al. [18]	Pathology (all patients)	Pathology/ Imaging FU	Pathology/ Imaging FU	Pathology/ Imaging FU
**2021**	Kuten J. et al. [19]	Pathology (all patients)	Pathology/ Imaging FU	Pathology/ Imaging FU	Pathology/ Imaging FU

n.p. analyses not performed in the study; *-* no reported reference standard.

**Table 4 diagnostics-12-01958-t004:** Yield FAPI vs. FDG PET and/or CECT/MRI; patient-wise and lesion-wise.

Year	Author	Type of Cancer	PATIENT-WISE ANALYSIS Number of Patients with Positive Findings (T/N/M/PC) * on FAPI PET (Confirmed by Reference Standard)	PATIENT-WISE ANALYSIS Number of Patients with Positive Findings (T/N/M/PC) on FDG PET and/or CECT/MRI (Confirmed by Reference Standard)	LESION-WISE ANALYSIS Number of Detected Lesions (N/M/PC/NS) by FAPI PET (Confirmed by Reference Standard)	LESION-WISE ANALYSIS Number of Detected Lesions (N/M/PC/NS) by FDG PET and/or CECT/MRI (Confirmed by Reference Standard)
**2021**	Chen, H. et al. [10]	P	-	-	- / - / - / 15 (1)	- / - / - / 2 (1)
		G	-	-	- / - / - / 57 (12)	- / - / - / 15 (12)
		Ch	-	-	- / - / - / 10 (4)	- / - / - / 2 (4)
**2021**	Guo, W. et al. [11]	Ch	7 (7) / - / - / -	4 (7) / - / - / -	-	-
**2021**	Pang, Y. et al. [12]	G	11 (11) / - / - / -	4 (11) / - / - / -	-	-
**2021**	Zhao, L. et al. [17]	P G	- / - / - / 6 (6) - / - / - / 13 (13)	- / - / - / 4 (6) - / - / - / 7 (13)	-	-
**2021**	Qin, C. et al. [13]	G	14 (14) / 12 (12) / 13 (13) / 10 (10)	10 (14) / 10 (12) / 13 (13) / 4 (10)	45 (12) / 37 (13) / 42 (10) / - / -	33 (12) / 22 (13) / 14 (10) / - / -
**2021**	Rohrich, M. et al. [14]	P	-	n.p.	n.p.	n.p.
**2021**	Shi, X. et al. [15]	Ch	3 (3) / - / - / - /	3 (3) / - / - / -	4 (4) / - / - / - /	4 (4) / - / - / -
**2021**	Shi, X. et al. [16]	G Ch	- / - / - / - 2 (2) / - / - / -	n.p.	- / - / - / - - / - / - / -	n.p.
**2021**	Pang, Y. et al. [18]	P	26 (26) / - / - / -	19 (26) / - / - / -	45 (15) / 88 (17) / 77 (10) / -	23 (15) / 22 (17) / 33 (10) / -
**2021**	Kuten, J. et al. [19]	G	10 (10) / 2 (2) / - / 5 (5)	5 (10) / 2 (2) / - / 0 (5)	16 (2) / - / - / -	16 (2) / - / - / -

* T/N/M/PC/NS = primary tumour/lymph node metastasis/organ metastasis/peritoneal metastasis/not further specified. P = pancreatic carcinoma; G = gastric carcinoma; Ch = cholangiocarcinoma. Patient-wise = ability of FDG or FAPI to detect presence of primary tumour, nodal disease, organ- or peritoneal metastasis (total number of patients with positive lesions per stage in brackets). Lesion-wise = total amount of lesions detected for nodal metastasis, organ or peritoneal metastasis (total number of patients with positive findings per stage in brackets). - not reported for the diseases of interest separately. n.p. analyses not performed in the study.

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
