# Peer review of "FAPI PET versus FDG PET, CT or MRI for Staging Pancreatic-, Gastric- and Cholangiocarcinoma: Systematic Review and Head-to-Head Comparisons of Diagnostic Performances"

_diagnostics, 2022, doi:10.3390/diagnostics12081958_

Round 1
Reviewer 1 Report
Regarding the details of the manuscript, my comments are as follows:
1) As the authors noted as a limitation, some studies are retrospective studies analyzed from the same prospective study. Although not 100%, most of the patients enrolled in these studies are bound to overlap. I believe that the same results are continuously listed and explained in this manuscript. Please exclude duplicate research results and re-describe appropriately.
2) The authors did not present “sensitivity” in the lesion-wide analysis in Table 6. Sensitivity analysis does not target only true positive lesions, and it is not understandable to explain that analysis was not performed because there was no biopsy or follow-up imaging. Please add why the sensitivity was not measured, or describe the sensitivity result.
3) The description of “n.p.*” in Table 7 is omitted. Please add abbreviations
4) Reference #14 and #16 are judged to be articles comparing FAPI PET with CR or MRI rather than FDG PET. In the result analysis, it should be added how the diagnostic performance was when the tumor was staged with CT or MRI. It is insufficient to explain that there is no FDG PET result on the result table because FDG PET is not performed.
5) Among the analysis of reference #13 in Table 7, the SUVmax of LNs in the patient-wise analysis was 9.9, whereas SUVmax of LNs in the lesion-wise analysis was described as 8.7. What is this difference?
6) Although the title of this review article is to analyze the difference between FAPI PET and FDG PET, CT, and MRI, the main content is considered to be a comparative analysis between FAPI PET and FDG PET. The proportion of parts analyzed for CT and MRI is small, so there seems to be no reason to add CT and MRI to the title. If the authors want to compare MRI and CT by adding them to the title, additional analysis and discussion of how lesions were diagnosed in CT and MRI are required.
Reviewer 2 Report
This is a comparative review of the use of FAPI-PET for diagnostic imaging of pancreatic, gastric, and cholangiocarcinoma, as compared to other more standard medical imaging methods (FDG PET/CT/MRI). Ten published studies (all published last year) were included in the analysis. This is a timely article addressing an important current topic of cancer imaging research. The authors have done a reasonable job of compiling the results of the recent studies. However, the study does seem a bit overenthusiastic in its conclusions, as there are clearly some important limitations, which are somewhat obscured and not adequately addressed in the present form of this manuscript.
1. Usefulness of FAPI-PET may be limited by specificity, which I feel is not adequately discussed. Intro states that FAP “expression levels in normal human tissues are generally very low”. However, FAP expression is clearly not restricted to cancer. The literature shows that several non-cancerous inflammatory conditions are associated with increased FAP expression, even in the organs of interest of this study. These include stellate cell activation in NASH / hepatitis, pancreatitis, etc. Some citations/discussions of these points are warranted. This is in addition to the issue of normal background uptake, which may be high in some organs (as acknowledged). Some of the inflammatory conditions are risk factors for cancer, complicating things further. This all casts doubt on the statement in the Discussion that non-cancerous FAPI-avid “lesions… would not be confused with potential primary or metastatic lesion of the three tumor types studied here”, especially considering the limited number of studies so far and very limited amount of histological / follow-up data. Intro and Discussion sections should be re-written to scale back the overall conclusions.
2. The study states that a median of 32 patients were included per study (Section 3.1). However, the median number of patients per study based on Table 2 is apparently only 27 patients. How did the authors arrive at a median of 32 patients per study?
